# How Close Is Existing C/C++ Code to a Safe Subset?

Christian DeLozier 

United States Naval Academy, 105 Maryland Ave, Annapolis, MD 21402, USA; delozier@usna.edu

**Abstract:** Using a safe subset of C++ is a promising direction for increasing the safety of the programming language while maintaining its performance and productivity. In this paper, we examine how close existing C/C++ code is to conforming to a safe subset of C++. We examine the rules presented in existing safe C/C++ standards and safe C/C++ subsets. We analyze the code characteristics of 5.8 million code samples from the Exebench benchmark suite, two C/C++ benchmark suites, and five modern C++ applications using a static analysis tool. We find that raw pointers, unsafe casts, and unsafe library functions are used in both C/C++ code at large and in modern C++ applications. In general, C/C++ code at large does not differ much from modern C++ code, and continued work will be required to transition from existing C/C++ code to a safe subset of C++.

**Keywords:** programming languages; safe subset; memory safety; C++

## 1. Introduction

For decades, the lack of memory safety in C and C++ has been the culprit behind a significant number of software vulnerabilities [1]. C allows for unchecked access to memory through pointers and arrays, and C++, its predecessor, maintains the same capabilities. These features make C and C++ powerful languages, giving programmers direct control over memory layout, allocation, and access. However, programmers often make mistakes involving memory, and these mistakes can lead to security vulnerabilities. Despite the numerous available approaches for preventing memory safety errors in these languages, developers have been hesitant to adopt any of them.

Following the rise in popularity of languages like Rust, multiple organizations have advised that software developers should transition away from using languages like C and C++ in favor of memory-safe languages [2,3]. In response, the C++ Standards Committee released a report on the future direction for ISO C++ [4]. As part of their response, the committee indicated that moving toward defining a *safe subset* of C++ could serve the purpose of allowing applications to be written in C++, thus gaining the advantages of C++'s performance and productivity while improving the safety of code written in C++. A safe subset of C/C++ restricts or bans the use of unsafe language features and, when necessary for productivity or performance, replaces unsafe language features with safer variants that accomplish the same goal. Prior work [5–8] has aimed to design a safe subset of C/C++. These efforts have generally restricted the use of types and operations that may allow unsafe accesses to memory when used incorrectly, and they have statically restricted the use of features outside of the safe subset via a compiler or static analyzer.

Our goal is to understand how close existing code is to conforming to a safe subset and to highlight the work that would be required to transition from existing C/C++ code to a safe subset of C++. We examine code that is predominantly C, predominantly C++, and a mix of both to investigate how the work required to conform to a safe subset might vary between code bases. To accomplish this task, we applied a static analysis tool that identifies potentially problematic code constructs in existing C/C++ code. We ran this static analysis tool on 5.8 million code samples from the Exebench benchmark suite [9], two C/C++ benchmark suites (LLVMTest and SPEC 2017) [10,11], and five modern C++ applications [12–16]. We analyzed the data to determine (Q1) how often raw pointers, void

pointers, and smart pointers are used in existing C/C++ code, (Q2) how commonly unsafe constructs are used in existing C/C++ code, and (Q3) how much "modern" C++ code differs, in terms of using unsafe constructs, from C/C++ code at large.

This paper makes three main contributions.

- We summarize the existing work on safe C/C++ standards and the safe subsets of C/C++.
- We present a static analysis tool and methodology for identifying potentially problematic code constructs in the existing C/C++ code.
- We analyze the data from 5.8 million code samples in the Exebench benchmark suite, two C/C++ benchmark suites, and five modern C++ applications to determine how close existing C/C++ code is to conforming to a safe subset of C++.

The remainder of the paper is organized as follows. Section 2 provides background information on the recent arguments over the safety of C/C++ and the motivation to transition toward a safe subset of C++. Section 3 analyzes and compares a set of existing safe C/C++ standards. Section 4 discusses the prior work on safe subsets of C/C++. Section 5 introduces our experimental methodology and provides background information on the Exebench benchmark suite, the LLVMTest and SPEC 2017 benchmark suites, and the modern C++ applications studied in the remainder of the paper. Section 6 presents the results of our experiments and discussion on the state of existing C/C++ code in relation to a safe subset of C++. Section 7 discusses the limitations of our study and potential for future work, and Section 8 concludes the paper.

## 2. Background

In recent years, there has been a growing chorus of concerns, primarily centered on issues related to software safety, surrounding the use of C and C++ programming languages. These apprehensions have arisen as a response to years of security vulnerabilities and incidents that have underscored the risks inherent in these languages. In response to these apprehensions, the software development community has actively sought ways to address safety concerns, resulting in the emergence of safe C++ standards and safe subsets of C++. This background section will delve into the contemporary arguments against using C and C++ due to safety concerns, explore the various safe C++ standards that have been developed, and examine the concept of safe subsets of C++, thereby shedding light on the ongoing efforts to strike a balance between the power and efficiency of these languages and the imperative of ensuring software security.

### 2.1. Memory Safety Errors in C/C++

C and C++ have long been plagued with memory safety errors due to the lack of an enforced type of safety, lack of bounds checking, and the use of manual memory management. These design choices make these languages more efficient in terms of performance, but they leave room for bugs that can be exploited as vulnerabilities.

Spatial errors occur when a program writes data past the boundaries of an allocated memory buffer. This can lead to corruption of adjacent memory locations and potentially result in security vulnerabilities. Such errors are often exploited by attackers to execute arbitrary code or gain unauthorized access to a system. Similar errors can occur when casts between pointer types allow for a write outside of an allocated object or a mismatch between the types being written to and used. Figure 1 demonstrates a buffer overflow similar to a vulnerability found in the SSH1 protocol [17]. In this code, an array is allocated with a size that is dependent on user input. If the user inputs a large enough number, the calculation may wrap around from a large number back to zero, thus leading to an attempt to allocate a zero-sized array. The subsequent access to the zero-sized array will write to memory that has not been allocated, which is a spatial safety error. In the context of a larger code base, errors like this can become security vulnerabilities.

```
unsigned int size = getInput();
int *values = new int[size + 8];
values[0] = 15;
```

**Figure 1.** Buffer overflow in C++ due to integer overflow. If the size calculation wraps around to 0, the array is allocated with size 0, thus leading to an overflow (even at index 0).

Lifetime errors occur when a program accesses a memory location after the associated object has been deallocated or when the pointer points to an invalid memory address. Figure 2 demonstrates a lifetime error caused by storing the address of a stack-allocated variable into a global pointer. The global pointer has a longer lifetime than the stack-allocated variable, thereby allowing the pointer to refer to the memory after it has been reallocated for a new purpose. These errors can lead to crashes, data corruption, and unpredictable behavior in a program. The manual memory management in C and C++ means that developers must be meticulous in tracking and managing pointers, which can be error-prone and complex. A number of similar issues, including null and uninitialized pointer dereferences, fall into the same category of errors.

```
int *p = nullptr;

void foo(){
  int x = 5;
  ...
  p = &x;
}

void bar(){
  int y = 0;
  ...
  *p = 10;
}
```

**Figure 2.** Use-after-free errors in C++ due to assigning a stack-allocated address into a global pointer.

Race conditions can exacerbate both of these issues. A lack of proper synchronization in concurrent code can lead to issues with uninitialized memory, null pointer dereferences, use-after-free errors, and buffer overflows. These bugs can be difficult to find and fix in parallel programs.

## 2.2. Recent Arguments against C/C++

In recent history, both the National Security Agency (NSA) and the National Institute for Standards and Technology (NIST) of the United States of America have published documents arguing against using C and C++. We summarize these arguments here for context on why safe coding standards and subsets have risen as a potential solution.

### 2.2.1. Executive Order (EO) 14028

In May 2021, Executive Order 14028 from the President of the United States mandated that NIST, operating in consultation with other relevant government agencies, publish guidance "identifying practices that enhance the security of the software supply chain [3]". In particular, Section 4(e)(ix) mandates guidance on "attesting to conformity with secure software development practices", and Section 4(r) mandates guidance on "recommending minimum standards for vendors' testing of their software source code, including identifying

recommended types of manual or automated testing". In response to this executive order, both the NSA and NIST released guidance that, in part, recommended the use of memory-safe languages.

### 2.2.2. NSA

On November 2022, the National Security Agency published a "Cybersecurity Information Sheet" focused on memory safety issues in unsafe languages like C and C++ [2]. This paper highlighted recent reports from Microsoft [18] and Google [19] regarding memory safety vulnerabilities in their products. As noted by the NSA, memory safety vulnerabilities are often exploited by attackers to gain remote code execution capabilities.

The NSA paper highlighted four issues with memory safety that can lead to vulnerabilities: buffer overflows, use-after-free errors, uninitialized variables, and race conditions. These types of bugs can occur in C and C++ programs due to the lack of array bounds checking, the use of manual memory management, a lack of requirements in initializing memory, and use of a weak consistency memory model for concurrency. Memory-safe languages solve these issues through static restrictions and dynamic checks. However, as noted by the NSA, "memory safety can be costly in performance and flexibility".

This paper goes on to discuss various approaches to defending against memory safety vulnerabilities, including static and dynamic analyses. Static analyses can be costly in terms of programmer flexibility and time, and dynamic analyses can negatively impact run-time performance. Other approaches offer band-aid solutions in the form of anti-exploitation features, but these can often be bypassed by a clever attacker.

The NSA paper concluded that the "path forward" should be to shift from using languages like C and C++ to memory-safe languages when possible.

### 2.2.3. NIST

In October 2021, the National Institute for Standards and Technology (NIST) released a document entitled "Guidelines on Minimum Standards for Developer Verification of Software". Overall, this document recommends techniques for software verification, including static and dynamic analysis, and it does not directly recommend moving away from languages like C and C++. However, in Section 3.1 of the document, the NIST recommends using compile-time flags that enable run-time buffer overflow checking and other memory safety protections. In Section 3.2 of the document, the NIST recommends using tools [20–22] that enforce memory safety in C and C++.

### 2.3. ISO C++ Response

In response to the opinions from the NSA and NIST on C/C++, an ISO C++ committee published an opinion safety in ISO C++ [4]. This opinion outlined a number of tenets as to where the ISO C++ committee does not want to go in order to provide a safer future for C/C++ code. We summarize those tenets here for context on what a safe subset of C++ should look like in the eyes of the ISO C++ committee. A safe subset should not break backwards compatibility. It should not remove the ability to express powerful abstractions in C+ or eliminate C++'s productivity advantages. A safe subset should not rely on purely run-time checks and therefore impose performance overheads that negate the advantage of using C++. It should provide semantic compatibility across different environments. Finally, a safe subset should not be one size fits all, may not fix every problem in every instance, must allow for gradual or partial adoption, and must not freeze development of C++ in other directions.

The committee's direction motivates the need to examine existing C/C++ code to determine the feasibility of gradually moving toward a hypothetical safe subset of C++ that maintains the strengths and advantages of C++. Given the lack of a standardized safe subset of C++, we examine this in the context of existing safe C/C++ standards and existing safe subsets of C/C++.

### 3. Safe C/C++ Standards

Safe subsets of C/C++ refer to restricted or controlled portions of the C/C++ programming languages that are designed to reduce the likelihood of programming errors and work to improve code safety. These subsets are typically used in critical or safety critical systems, where robustness and reliability are paramount. By limiting certain features and adopting stricter rules, developers can mitigate the potential pitfalls and vulnerabilities associated with the full C/C++ language.

A number of organizations have proposed standards and guidelines for writing safe C and C++ code. We provide an overview of these standards and guidelines for context on the features of C and C++ that can be considered harmful to safety.

#### 3.1. CPP Core Guidelines

One of the most well-known safe subsets of C++ is "C++ Core Guidelines" [23]. These guidelines cover a wide range of topics, including memory management, exception handling, and design principles, all aimed at producing more maintainable and less error-prone code. The C++ Core Guidelines are organized into the categories of philosophy (P), interfaces (I), functions (F), classes and class hierarchies (C), enumerations (Enum), resource management (R), expressions and statements (ES), performance (Per), concurrency and parallelism (CP), error handling (E), constants and immutability (Con), templates and generic programming (T), C-style programming (CPL), source files (SF), and the standard library (SL).

The C++ Core Guidelines also include the C+ Core Profiles for type safety, bounds safety, and lifetime safety, which provide additional guidance on writing C++ code that preserves these important properties. The type safety profile recommends recognizing avoiding casts, using `dynamic_cast` to downcast, avoiding C-style casts, initializing all variables, avoiding unions, and avoiding `varargs`. The bounds safety profile recommends avoiding pointer arithmetic, only indexing into arrays with constant expressions, avoiding array to pointer decay, and avoiding unsafe library functions. Finally, the lifetime safety profile recommends avoiding dereferencing potentially invalid pointers.

#### 3.2. MISRA C++

In safety critical industries such as the aerospace and automotive industries, MISRA (Motor Industry Software Reliability Association) C++ is a widely used safe subset of the language [24]. MISRA C++ is derived from the MISRA C guidelines and extends the same principles to C++. It focuses on avoiding undefined behavior, reducing the use of dangerous language features, and promoting a consistent coding style to enhance code readability and maintainability. The MISRA C++ rules cover the categories of language-independent issues, general, lexical conventions, basic concepts, standard conversions, expressions, statements, declarations, declarators, classes, derived classes, member access control, special member functions, templates, exception handling, preprocessing directives, libraries, language support libraries, diagnostics libraries, and input/output libraries.

#### 3.3. AUTOSAR C++

AUTOSAR (Automotive Open System Architecture) C++ is a coding standard specifically tailored for the automotive software industry, and it is intended to ensure the development of safe and reliable C++ code for embedded systems in vehicles [25]. It defines guidelines and rules for C++ programming, with an emphasis on adhering to the principles of AUTOSAR architecture and supporting automotive safety standards. AUTOSAR C++ promotes consistency, maintainability, and reliability in automotive software development by providing specific coding practices, naming conventions, and design considerations aligned with the AUTOSAR framework. AUTOSAR rules are classified as mandatory (M) or advisory (A), and these rules may overlap as requirements and recommendations. AUTOSAR rules are also classified based on how easy the rule is to automatically enforce. Rules that cannot be automatically enforced are classified as non-automated. The AUTOSAR

rules cover the categories of language-independent issues, general, lexical conventions, basic concepts, standard conversions, expressions, statements, declarations, declarators, classes, derived classes, member access control, special member functions, overloading, templates, exception handling, and preprocessing directives.

### 3.4. CERT

The CERT (Computer Emergency Response Team) C/C++ Coding Standard, developed by the CERT Division of the Software Engineering Institute (SEI) at Carnegie Mellon University, is a set of guidelines and best practices for writing secure and reliable C and C++ code [26]. The standard is designed to help software developers and organizations reduce vulnerabilities and improve the overall quality of their code. The CERT coding standard consists of 11 categories of recommendations, including declarations and initialization (DCL), expressions (EXP), integers (INT), containers (CTR), characters and strings (STR), memory management (MEM), input output (FIO), exceptions and error handling (ERR), object-oriented programming (OOP), concurrency (CON), and miscellaneous (MSC).

### 3.5. High Integrity C++

The High Integrity C++ standard consists of coding rules and best practices for writing high-quality C++ code [27]. The High Integrity C++ standard has 155 rules that cover various aspects of C++. The HIC standard is categorized into numeric divisions as follows: (1) general, (2) lexical conventions, (3) basic concepts, (4) standard conversions, (5) expressions, (6) statements, (7) declarations, (8) definitions, (9) classes, (10) derived classes, (11) member access control, (12) special member functions, (13) overloading, (14) templates, (15) exception handling, (16) preprocessing, (17) standard library, and (18) concurrency.

### 3.6. Joint Strike Fighter

The Joint Strike Fighter (JSF) C++ Coding Standard was developed by Lockheed Martin for work on the F-35 fighter jet [28]. This coding standard attempts to provide direction and guidance to C++ programmers that lead to safe, reliable, testable, and maintainable C++ code. The JSF AV (air vehicle) standard draws on prior rule sets, including MISRA [24], the Vehicle Systems Safety Critical Coding Standards for C, and the C++ language-specific guidelines and standards. The JSF rules are classified as *should*, *will*, and *shall* rules. The *should* rules are recommended but not required. The *will* rules are mandatory but do not require verification. The *shall* rules are mandatory and must be verified. The JSF coding standard rules cover the following language features: environment, libraries, pre-processing directives, header files, implementation files, style, classes, namespaces, templates, functions, comments, declarations and definitions, initialization, types, constants, variables, unions and bit fields, operators, pointers and references, type conversions, flow control structures, expressions, memory allocation, fault handling, portable code, efficiency considerations, and miscellaneous.

### 3.7. Summary and Comparison

Table 1 highlights the rules in these C/C++ standards that correspond to memory safety properties. Unlisted rules generally fall into the categories of code style, cleanliness, maintainability, and performance. Of course, many of these rules, especially those that involve maintainability and cleanliness, may be related to memory safety issues. The rule below provides one such example.

**C++ Core F.8:** Prefer pure functions.

Rules like this improve the maintainability, readability, and cleanliness of code. Lifetime errors are less likely in functions with no side effects. Likewise, preventing spatial safety errors may be easier in code without side effects.

Other rules, especially those relating to spatial and lifetime safety, may be difficult to enforce, as acknowledged by the standards themselves.

**CERT EXP54-CPP:** Do not access an object outside its lifetime.

**JSF AV 70.1:** An object shall not be improperly used before its lifetime begins or after its lifetime ends.

**AUTOSAR M5-2-5:** An array or container shall not be accessed beyond its range.

Multiple standards advocate for some form of run-time checks to ensure that that safety requirements are met. The implementation of run-time checks is left up to some combination of the programmer and tool chain.

**JSF AV 15:** Provision shall be made for run-time checking (defensive programming).

**CERT STR53-CPP:** Range check element access.

Following a C/C++ standard will improve the quality of C/C++ code, but security vulnerabilities like buffer overflows and lifetime errors cannot be prevented exclusively at compile time. A safe subset of C++ will likely need to rely on both the static enforcement of conformance to a subset and dynamic checking for run-time errors.

**Table 1.** Summary of the rules in Safe C++ Standards that directly relate to memory safety properties.

| Safe C/C++ Standard | Initialization | Spatial | Lifetime | Type |
|---|---|---|---|---|
| Core guidelines [23] | I.12, F.60, F.22, F.23, ES.20, ES.22 | I.13, C.90, C.152, R.14, ES.27, ES.42, ES.55, ES.71, ES.103 | I.11, F.7, F.26, F.27, F.42, F.43, F.44, F.45, F.53, C.21, C.31, C.33, C.49, C.82, C.127, C.149, R.1, R.3, R.4, R.5, R.10, R.11, R.36, ES.61, ES.65 | I.4, F.55, C.46, C.146, C.164, C.181, C.182, C.183, ES.34, ES.48, ES.49, SL.4 |
| MISRA [24] | 0-3-1, 8-5-1 | 0-3-1, 3-1-3, 5-0-15, 5-2-12, 18-0-5, 27-0-1 | 0-3-1, 7-5-1, 7-5-2, 7-5-3, 15-0-2, 18-4-1 | 5-2-2, 5-2-4, 5-2-6, 5-2-7, 5-2-8, 9-5-1 |
| Autosar [25] | A3-3-2, A5-3-2, A8-5-0, A12-6-1 | M5-0-15, M5-2-5, M5-2-12, A17-1-1, M18-0-5, A18-1-1, A27-0-4, A27-0-2 | A3-8-1, A5-1-4, A5-3-3, M7-5-1, M7-5-2, A7-5-1, A8-4-11, A8-4-12, A8-4-13, A18-1-4, A18-5-1, A18-5-3, A18-5-8, A20-8-1 | M5-2-2, A5-2-1, A5-2-2, A5-2-4, M5-2-6, M5-2-8, M5-2-9, A9-5-1, A13-5-2, A13-5-3 |
| CERT [26] | EXP53 | CNTR50, CTR53, CTR55, STR50, STR53 | EXP51, EXP54, EXP61 | DCL50, EXP58, INT50, MEM50, MEM51, MEM56 |
| HIC [27] | 8.4.1 | 4.1.1, 6.2.1, 17.2.1 | 3.4.1, 3.4.2, 8.1.1 | 3.5.1, 5.4.1, 5.4.3, 12.1.1 |
| JSF [28] | 71, 71.1, 117, 118, 142, 143, 174 | 15, 20–25, 96, 97, 215 | 70.1, 111, 173, 206 | 153, 178, 179, 182, 183, 185 |

## 4. Safe Subsets of C/C++

Safe subsets of C/C++ take safety one step beyond secure coding standards by replacing language features that are difficult to ensure safety for with safe constructs and banning language features that cannot reasonably be secured. We highlight three safe subsets of C and one safe subset of C++ from prior work.

## 4.1. SafeC

SafeC [5] improves the memory safety of the C programming language by replacing pointers with a SafePtr structure that enables memory safety checks. The SafePtr structure includes the original pointer, a base pointer, a size, a storage class (heap, local, and global), and a capability.

To apply SafeC to existing C code, all pointers must be translated to a safe pointer structure. Once pointers have been translated, safety checks can be inserted into the code that perform tests using the additional metadata. Operations on pointers must also be modified to produce new pointer structures.

SafeC introduced the concept of securing C through transformations on pointers, but its implementation was limited by the lack of template support in C++ at that time. The safe pointer definition has a template-like definition, but it cannot not use operator overloading to implement the safety checks or pointer operations. SafeC highlights the necessity of securing pointer operations to produce secure code written in C.

## 4.2. CCured

Similar to SafeC, CCured [6] adds memory safety guarantees to C programs via program transformations on pointers. CCured introduces *SAFE*, *SEQ*, *WILD*, and *RTTI* pointer types to replace raw pointers in C programs. *SAFE* pointers do not use pointer arithmetic or casts. *SEQ* pointers can use pointer arithmetic but not casts. *WILD* pointers can perform both pointer arithmetic and casts. *SEQ* and *WILD* pointers must carry additional metadata to secure these additional operations. *RTTI* pointers allow downcasts by carrying run-time type information. CCured's automatic program transformation identifies, via a pointer analysis, the correct type for each pointer in a program.

In a real-world study [29], the authors of CCured examined the types of pointers that were required to secure C programs. In a set of Apache models, they found that the expensive *WILD* and *RTTI* pointers were rarely required. Similarly, in a set of system software applications, the *SAFE* and *SEQ* pointers made up the majority of pointers required. This study emphasizes the importance of identifying how pointers are used at the source level of a C program to efficiently enforce memory safety.

## 4.3. Cyclone

Cyclone [7] restricts unsafe idioms in C and provides extensions to allow programmers to safely use similar constructs. Cyclone requires that pointers are initialized before use and applies safety checks to pointer usage. Cyclone restricts casts and provides tagged unions to ensure type safety. Cyclone restricts the use of control-flow constructs like `goto`, `setjmp`, and `longjmp`. Cyclone introduces the idea of using static analysis in a compiler to ensure that the code conforms to the safe subset of C.

## 4.4. Ironclad C++

Ironclad C++ [8] implements a safe subset of C++ by replacing raw pointers with templated smart pointers. An automatic refactoring tool replaces pointer types in C/C++ code with smart pointers, and the smart pointer classes implement safety checks on operations using operator overloading. A static analysis tool ensures that programs conform to the safe subset. Like prior work, Ironclad C++ introduces pointer types with different capabilities and required checks. The `ptr` class allows only singleton pointers to heap or global memory, and the `aptr` class allows pointers to arrays with the same storage classes. The `lptr` and `laptr` classes allow pointers to stack-allocated memory and implement a safety check on stack scoping.

Ironclad C++ restricts the use of unsafe idioms in C/C++ through the use of a static analysis validator. Pointer types must be replaced by smart pointers, and all casts must use the `cast` template, which enforces run-time type checking. Unions are disallowed. References cannot be class members, and references cannot be initialized from the dereference of a pointer. These rules prevent dangling references and are required because reference

operations cannot be effectively wrapped by Ironclad C++ due to the lack of operator overloading for the dot operator.

Ironclad C++ identified a number of corner cases in which it was difficult to translate from existing C/C++ code to a safe subset of C++. Variable names in C, such as `this`, came into conflict with C++ keywords. Refactoring unsafe casts often required programmer intervention, especially in situations that required a class hierarchy to replace C-style "classes" using void pointers. Additional effort was also required to translate code that relied on implicit casts from string literals to character pointers in a constructor because replacing the raw character pointer with a smart pointer in a constructor exceeds the number of implicit casts allowed by the C++ compiler.

SaferCPlusPlus [30] is a more thorough implementation of Ironclad C++ that wraps many of the library features provided by C/C++ to ensure safety.

### 4.5. What Work Has Been Required to Translate to Previous Safe Subsets of C++?

Moving from standard C/C++ code to a safe subset of C++ would likely require translating unsafe constructs to safe constructs. Generally, this task can be accomplished by replacing unsafe constructs in the source code or instrumenting the unsafe code with a compiler extension. For this analysis, we focused on translation at the source code level because it better captured the workload required to move from unsafe C/C++ code to a safe subset. We also noted that compiler transformations may have difficulty capturing source-level semantics such as singleton versus array pointers and class hierarchies that can impact the performance of the translated code. We assumed that backwards compatibility is a requirement for any safe subset, and as such did not assume that existing language constructs can be co-opted into a new functionality.

Given that pointers are the main source of unsafety in C/C++, moving to a safe subset will require replacing raw pointers with types that can provide capabilities such as null checking, bounds checking, and lifetime checking when necessary. Prior work [8,29] has shown that the run-time performance overhead of safety checks on pointers can be reduced by matching pointer types to their required capabilities. With smart pointer classes, or potentially a new set of basic types for pointers, only the types of pointers would need to be translated to move to a safe subset. Operations on pointers can use the same syntax as they do now.

Functions that create pointers, such as `malloc` and `new`, would also need to be modified or translated. New allocation functions would create new pointer types that carry the required metadata for safety checks. Likewise, operations like address-of (`&`) may need special treatment to ensure that the metadata is maintained. Unsafe functions, like `strcpy`, must be replaced with safe versions of those functions.

Unions and unchecked casts would also need to be eliminated or replaced. In general, treating one type as another type should be checked at run time or proven at compile time to be safe.

Existing code may feature language constructs that are difficult to automatically translate to a safe subset of C++, and they would therefore require more programmer effort to realize a safe subset. Replacing raw pointers with smart pointers can cause more implicit casts to be required than the number that is allowed to be used by the C++ language. C++ allows an implicit conversion sequence of up to one standard conversion, up to one user-defined conversion, and up to one standard conversion after the user-defined conversion [31]. A standard conversion consists of up to one lvalue-to-rvalue, array-to-pointer, or function-to-pointer conversion, as well as up to one numeric conversion, up to one function pointer conversion, and up to one qualification conversion. The limit of one user-defined conversion may be an issue for smart pointer replacements for raw pointers because the constructor for the smart pointer is a user-defined conversion; as such, a class that takes a pointer as one of the parameters to its constructor may now need two implicit user-defined conversions. Figure 3 demonstrates this issue. In this example, the `Slice` class has a constructor that takes a `char*` as an argument. This constructor allows a `Slice` to be

implicitly constructed from a string literal to match a function or method definition that takes a `Slice` as a parameter. If we introduce a smart pointer class and wrap the `char*` as a `smart_pointer<char>`, the code in the if statement conditional would now require two user-defined casts. One of these casts would need to be made explicit in the source code. Generally, this issue can be mitigated by either applying an explicit conversion or using a first-class type instead of a smart pointer class to replace raw pointers.

```cpp
class Slice{
  char *_str;
  Slice(char *str) : _str(str) {}
  bool equals(Slice & other) {
    ...
  }
};

void CompareSlices(){
  Slice s1("OK"); // Constructor is explicit
  // Implicit conversion from StringLiteral
  if (s1.equals("NOT")) {
    ...
  }
}
```

**Figure 3.** An example of code that may be impacted by introducing smart pointers due to the limited number of user-defined casts that can be performed per implicit conversion sequence.

References have also posed a challenging issue for previous safe subsets because the dot operator cannot be overloaded. However, references are not as difficult to secure as pointers because the address that a reference refers to cannot be changed after the reference has been initialized. Therefore, references can partially be secured by static rules rather than via run-time enforcement. Ironclad C++ [8] secures references by disallowing reference class members and restricting the values that could be used as a return value by reference. In particular, only the dereference of a smart pointer, a reference function parameter, the dereference of the `this` pointer, or a class member could be returned by reference from a method. Existing safe C++ standards also restrict the values that can be returned by reference and generally restrict how references can be used in C++ code.

Array operations, especially on two-dimensional or higher arrays, can be difficult to implement efficiently when bounds checks are required for safety. In C/C++, two-dimensional arrays are naturally represented as a pointer to a pointer. Accessing the elements of a two-dimensional array therefore requires two pointer dereferences, which may require two safety checks. This issue can be mitigated by using a large one-dimensional array to represent a two-dimensional array, but fixing the issue in this manner negates the productivity benefit of representing the two-dimensional array naturally. Taking the address of an array element can also be problematic for a safe subset of C++ because the address of an operator produces a pointer into the array that may lack the supporting metadata from the original array. Most existing safe C/C++ standards restrict code to be used only in a single pointer indirection and disallow multiple levels of indirection.

*4.6. Summary*

Safe subsets have been previously applied to both C and C++ code. SafeC, CCured, and Cyclone demonstrated that C code could be secured using a safe subset, but the implementation used in these approaches was limited by the lack of class features like operator overloading and templates in C. Ironclad C++ and SaferCPlusPlus demonstrated how both C/C++ code could be secured using smart pointers and function wrappers.

In a discussion of a safe subset of C++, the goal of securing C, C++, or a mix of the two languages must be considered. In our view, it is difficult to separate C++ from the unsafe features of its predecessor, including raw pointers, arrays, unchecked casts, and unsafe functions from the C library. Therefore, we examined applications that are predominantly C, a mix of C and C++, and predominantly C++ in this study to fully understand the effort required to refactor these types of applications to a safe subset of C++.

## 5. Methodology

We developed a static analysis tool to identify the relevant features in C/C++ code and applied that tool to all of the code samples in the Exebench benchmark suite. We also applied the same tool to a set of larger, modern C++ applications. Our goal in using this static analysis tool was to answer three main experimental questions.

(Q1) How often are raw pointers, void pointers, and smart pointers found in existing C/C++ code?
(Q2) How often are "problematic" code constructs found in existing C/C++ code?
(Q3) Is "modern" C++ code closer to a safe subset than C/C++ code at large?

*5.1. Static Analysis Patterns*

In this section, we described the static analysis tool that we used to identify language features in C/C++. This tool was developed using LLVM and clang version 17.0.0. This tool is open source and can be found on Github (https://github.com/crdelozier/subsets accessed on 22 December 2023). We used clang's ASTMatchers library [32] to identify the patterns matching the language features described in the previous section. We highlighted the relevant rules from existing C++ standards that motivate the static analysis patterns that we chose to identify in the source code.

Figure 4 provides an example of one of the matcher rules used in this study. As shown, nodes in the abstract syntax tree can be identified by name and type, and quantifiers can be used to match against one or more specified patterns. Each matcher rule is paired with a `MatchCallback` that allows further processing of the matched AST nodes.

```cpp
StatementMatcher UnsafeCastMatcher =
    anyOf(
        cStyleCastExpr().bind("cast"),
        cxxReinterpretCastExpr().bind("cast")
    );

class UnsafeCastFinder : public MatchFinder::MatchCallback {
public :
    virtual void run(const MatchFinder::MatchResult &Result) override {
        ASTContext *Context = Result.Context;
        const CastExpr *VD = Result.Nodes.getNodeAs<CastExpr>("cast");
        // ...
    }
};
```

**Figure 4.** Example of a static analysis pattern using the ASTMatchers library. The bind() operation allows the MatchCallback to access the AST nodes that matched the pattern.

5.1.1. Pointers

As a first step, the static analysis tool identifies all the pointers, void pointers, and smart pointers in the C/C++ code. Enforcing safety on pointers is a critical task to move

toward a safe subset, and work will likely be required by programmers to ensure that pointers are used safely. This need to understand how often pointers are used in existing code is motivated by the existing rules in safe C++ standards and subsets.

**C++ Core ES.42:** Keep use of pointers simple and straightforward.

**C++ Core ES.65:** Do not dereference an invalid pointer.

**C++ Core R.3:** A raw pointer is non-owning.

**C++ Core I.11:** Never transfer ownership by a raw pointer (T*) or reference (T&).

*Pointers* are identified as any declaration with `PointerType`. On each pointer match, the tool checks for Void Pointers, which are identified by the `VoidPointerType`.

Smart Pointers are identified by searching for declarations with a C++ class type with the names `unique_ptr`, `shared_ptr`, `weak_ptr`, `auto_ptr`, and `ptr`. We note that `ptr` is not a standard library smart pointer type, but at least one of the applications that we examined created their own smart pointer wrappers with this name.

5.1.2. Unsafe Functions

Unsafe functions have long been a problematic source of errors in C/C++ code. Migrating a safe subset of C++ will likely require replacing uses of unsafe functions or wrapping them, and moving from C to C++ will require replacing uses of malloc and free. This is motivated by existing rules in the safe standards of C/C++.

**C++ Core R.10:** Avoid malloc() and free().

**AUTOSAR A18-5-1:** Functions malloc, calloc, realloc, and free shall not be used.

**C++ Core SL.4:** Use the standard library in a type-safe manner.

**HIC 17.2.1:** Wrap the use of the C standard library.

**AUTOSAR A17-1-1:** Use of the C standard library shall be encapsulated and isolated.

Unsafe Functions are identified by first matching all call expressions with Pointer arguments. The called function name is then compared against a list of known unsafe functions, such as `memcpy`, textttstrcmp, and `puts`. Calls to Malloc and Free are also identified by function name.

5.1.3. Casts

Unchecked casts can break type safety in C/C++ code. Existing C/C++ standards highlight the need to avoid C-style casts and other unchecked casts.

**C++ Core ES.48:** Avoid casts.

**C++ Core ES.49:** If you must use a cast, use a named cast.

**C++ Core C.146:** Use dynamic_cast where class hierarchy navigation is unavoidable.

**HIC 5.4.1:** Only use the casting forms static_cast (excl. void*), dynamic_cast, or explicit constructor call.

**AUTOSAR A5-2-2:** Traditional C-style casts shall not be used.

**AUTOSAR A5-2-4:** The reinterpret_cast shall not be used.

**MISRA 5-2-4:** C-style casts (other than void casts) and functional notation casts (other than explicit constructor calls) shall not be used.

As noted in the prior section, implicit casts to constructors may complicate the transition to a safe subset because smart pointers add an extra user-defined conversion. The C++ Core Guidelines also recommend avoiding implicit casts to constructors.

**C++ Core C.46:** By default, declare single-argument constructors as explicit.

Like unsafe casts, unions break type safety in C/C++ code by allowing an implicit conversion between types. C/C++ standards recommend avoiding unions and replacing them with tagged unions.

**C++ Core C.181:** Avoid "naked" unions.

Unsafe Casts are identified as C-style casts and `reinterpret_cast`. Construct from Implicit Casts are identified by finding implicit cast expressions with constructor ancestors in the abstract syntax tree. In this case, we use the ancestor matcher instead of the parent matcher because there may be multiple casts applied to a single constructor. Unions are identified by type.

### 5.1.4. References

References can pose a uniquely difficult problem to solve for a safe subset because they can cause similar lifetime and initialization errors to pointers, but they cannot be checked as effectively due to the inability to check for null references or to overload the dot operator. Existing C++ standards and subsets specify rules for how references should be used, especially in the context of return by reference.

**C++ Core F.43:** Never (directly or indirectly) return a pointer or reference to a local object.

**JSF AV 111:** A function shall not return a pointer or reference to a non-static local object.

**MISRA 7-5-1:** A function shall not return a reference or a pointer to an automatic variable (including parameters) defined within the function.

**MISRA 7-5-3:** A function shall not return a reference or pointer to a parameter that is passed by reference or a const reference.

Reference Class Members are identified as reference-type declarations with a class ancestor. Reference Returns and const Reference Returns are identified by finding function declarations that return a reference type, and which further identify the constant references. *Reference to Dereferenced* are identified as variable declarations with a reference type that are initialized with the pointer dereference unary operator.

### 5.1.5. Arrays

Of course, without bounds checking on array operations, a safe subset will be doomed. Out-of-bounds reads and writes are still some of the most common vulnerabilities found in C/C++ code [1]. We examined two issues with arrays that may affect a safe subset. First, array-to-pointer decay may lose information about the bounds of the array. Array-to-pointer decay can happen in multiple contexts, including when the address of an array element is taken. Bounds checking on arrays may also hinder the run-time performance of a safe subset if two-dimensional arrays are not handled carefully.

**C++ Core I.13:** Do not pass an array as a single pointer.

**C++ Core ES.27:** Use std::array or stack_array for arrays on the stack.

**HIC 8.1.1:** Do not use multiple levels of pointer indirection.

The Address of Array Subscript operations are identified by the address of operator with an array subscript operand, and 2D Arrays are identified as array subscript expressions with an array subscript expression as an ancestor. We note that this may also find further nested arrays, such as three-dimensional arrays.

### 5.2. Running Analysis on Exebench

ExeBench [9] is an innovative benchmark suite with the primary objective of expanding the potential of machine learning in the fields of compilation and software engineering. The data set addresses the critical challenge of limited available data sets for various tasks, such

as neural program synthesis and machine learning-guided program optimization. One of the key issues tackled by ExeBench is the scarcity of real-world code with references to external types and functions, as well as the need for the scalable generation of IO examples. It stands out as the first publicly accessible data set that pairs the authentic C code sourced from GitHub with IO examples, thereby enabling the execution of these programs. To accomplish this, a specialized tool chain was developed to scrape GitHub, analyze the code, and produce runnable code snippets. Exebench contains 5.8 million compilable functions from the Anghabench [33] and the Github public archive [34]. The code samples from Exebench are largely C code with a small amount of C++.

We developed a Python script to execute the static analysis tool on each code sample in Exebench. This tool extracts the C/C++ code samples from the Exebench JSON files and outputs a C++ file. This file includes both the function under test and the supporting function definitions, headers, and variables to allow the function to be compiled. We verified that the file can compile using *gcc/g++*. We passed the lines numbers of the function under test to the static analysis tool to isolate our analysis to that specific function to avoid including the synthetic test code generated by Exebench.

*5.3. Running Analysis on C/C++ Benchmark Suites and Modern C++ Programs*

We also ran our static analysis tool on a set of C/C++ benchmarks and modern C++ applications, as described in Table 2. We identified the modern C++ applications by reading threads about the projects that demonstrate modern C++ coding style on popular social media platforms and programming forums, including Reddit and HackerNews. We then downloaded the source code for these applications and ran the static analysis tool by replacing the `CC` and `CXX` build variables with a Python (version 3.7.0) wrapper script that runs the static analysis tool prior to running the standard compiler. This approach allowed us to gather data on these open-source applications without needing to manually run the static analysis tool on every C++ file in the project. We counted the lines of code in these applications using the `cloc` utility.

**Table 2.** Applications and code sample benchmark suites studied with the static analysis tool. Lines of code are measured using the Count Lines of Code (cloc) [35] tool.

| Application/Suite | Description | Language | Lines of Code |
|---|---|---|---|
| Cereal [12] | Serialization Library | C++ | 31,355 |
| Fmt [13] | Formatting library alternative | C++ | 43,993 |
| Folly [14] | Facebook core library components | C++ | 377,963 |
| Json [15] | JSON support for modern C++ | C++ | 103,137 |
| Redex [16] | Bytecode optimizer for Android | C++ | 281,642 |
| Exebench [9] | Exebench code samples | C | 112,710,884 |
| LLVM Test Suite [10] | LLVM Code Samples | C/C++ | 1,366,158 |
| SPEC CPU 2017 [11] | SPEC 2017 Benchmarks | C/C++ | 4,806,869 |

The LLVMTest benchmark suite [10] consists of a set of C/C++ programs that are used to benchmark clang and LLVM. The applications range from microbenchmarks and single-file applications to larger programs with multiple source files.

The SPEC 2017 benchmark suite [11] features a set of performance benchmarks for testing CPU performance. There are 9 predominantly C applications, 3 applications that

are a mix of C and C++, and 5 applications that are predominantly C++. Compared to the other applications studied, these benchmarks focus on computationally intensive tasks such as artificial intelligence, compression, and molecular dynamics. These applications tend to use more array operations compared to the other code samples in this study.

To avoid counting code constructs in header files multiple times, we included only statistics from the main file for these programs. We note that the main file includes templated classes from headers if the template has been instantiated in that file.

## 6. Results

We present the results collected by running the static analysis tool on 5.8 million code samples from Exebench, two C/C++ benchmark suites, and the modern C++ applications, as listed in Table 2.

### 6.1. Pointers, Smart Pointers, and Void Pointers

Figure 5 shows the percentage of pointer variables that are raw pointers, void pointers, and smart pointers. As shown, the large majority of pointer variables in all of the code samples examined were raw pointers. Void pointers are fairly common, even in modern C++ applications. Smart pointers are relatively rare. Clearly, quite a bit of work needs to be conducted to eliminate raw pointers from C/C++ code to transition to a safe subset that can provide checked pointer operations.

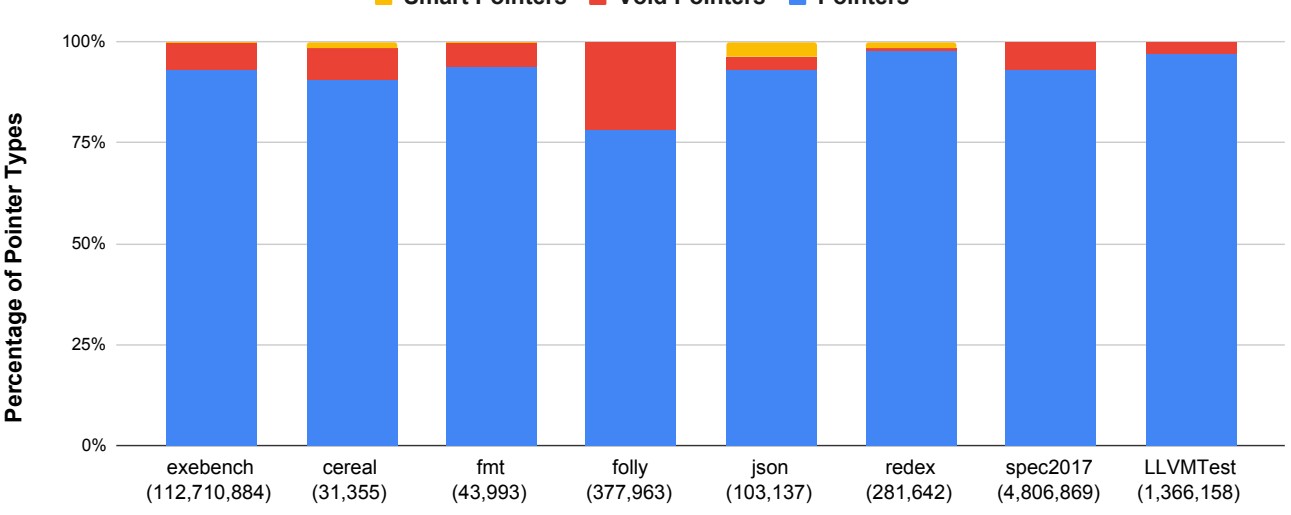

**Figure 5.** Pointers found in Exebench and other C++ applications. Each bar shows the percentage of all pointer variables that are raw pointers, void pointers, and smart pointers.

These results may be somewhat expected based on the existing safe C/C++ standards. Only the C++ Core Guidelines advocate strongly for smart pointer use, and its recommendations for using smart pointers are limited to certain allocation ownership scenarios. On the other hand, previous work on safe subsets of C/C++ have advocated that smart pointers, or other pointer type replacements, will be necessary to dynamically enforce memory safety properties.

Likewise, the results for void pointers may be expected based on existing standards. Void pointers are generally used to implement polymorphism without class hierarchies or to enable the storage of data with any type. We expect that the relatively large ratios of void pointers found in these modern C++ applications are used to enable the storage of data with any type, especially for use cases such as serialization. Existing standards caution against the use of `dynamic_cast`, which may hinder the performance and cause of unexpected

run-time errors when a cast fails. To avoid the use of `dynamic_cast`, programmers may err on the side of using void pointers to implement polymorphism.

### 6.2. Unsafe Functions

Figure 6 shows the total number of mallocs, frees, and unsafe library functions found in the C/C++ samples and applications. From the data, we can see that `malloc`, `free`, and other unsafe functions are still used even in some modern C++ applications, and they are prevalent in the Exebench function samples. We note that there are more frees than mallocs because multiple frees may be required per `malloc` to handle all the potential control flow paths along which an allocation must be freed. `redex` uses `jemalloc`, which does not match the function call name identifiers used in our tool, and so we identified no uses of `malloc` in `redex`, but we did identify uses of `free`.

We find that modern C++ code is still using unsafe library functions despite decades of recommendations to stop using them. In Exebench, 27% of all code samples use an unsafe library function. It is possible that these uses of unsafe library functions are wrapped as recommended by multiple C++ standards, but our static analysis tool cannot effectively check for this appropriate wrapping. Work will be required to replace or wrap uses of these unsafe library function calls to move toward a safe subset of C++. As for future work, we also intend to identify which unsafe functions are still widely used and why they are still used instead of safe alternatives.

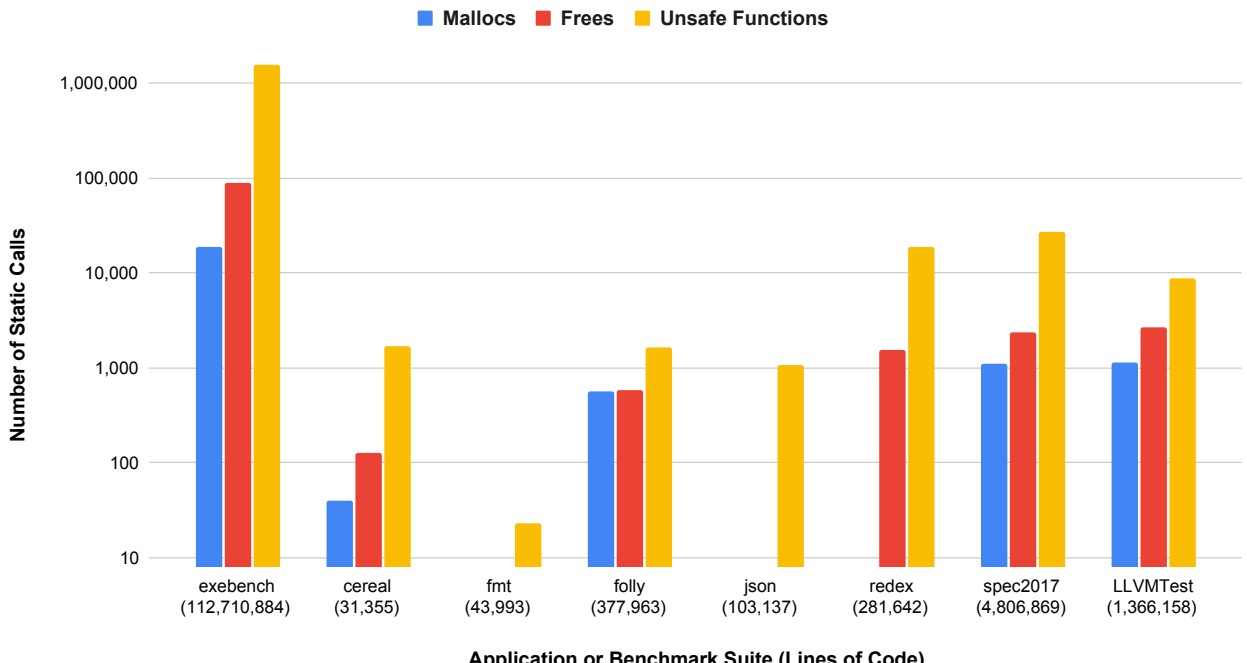

**Figure 6.** Calls to unsafe functions found in Exebench and other C++ applications.

Figure 7 shows a heat map of the unsafe function calls in `folly`. The heat map was generated by recording the source location of each unsafe function used in `folly`. We then mapped the file and line numbers to the bins between 0 and 99. The number of unsafe functions in each bin was normalized to values between 0 and 1.0 using NumPy (version 1.26.2), and the graphic was generated using MatPlotLib (version 3.8.2). This heat map demonstrates that the use of unsafe functions, at least in this application, were not concentrated in one particular file or even in one location in each file. Apart from a few relatively safe files, the uses of unsafe functions were spread throughout the application.

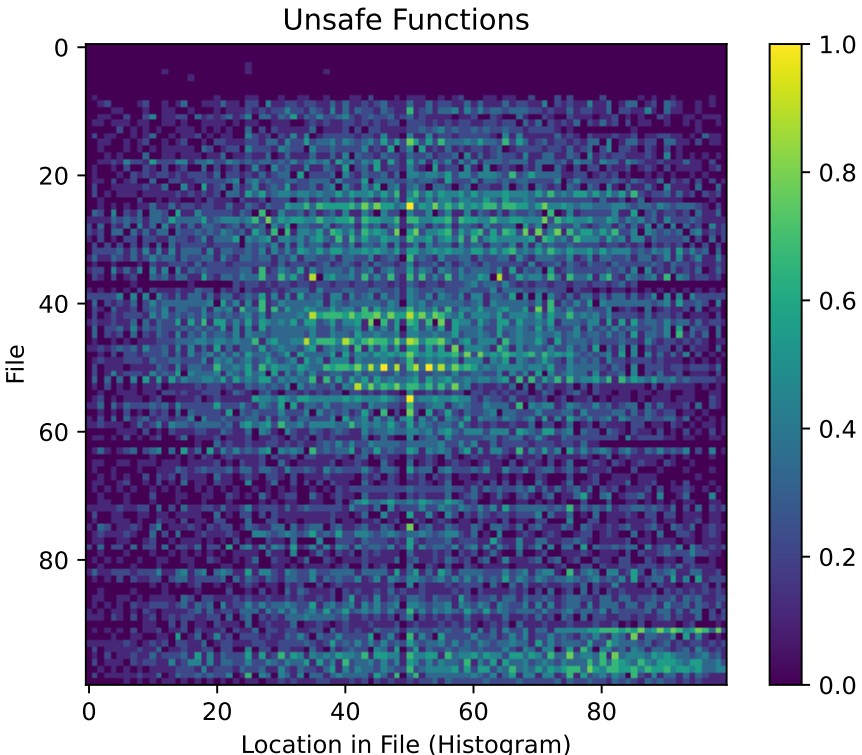

**Figure 7.** Heat map of calls to unsafe functions in `folly`.

### 6.3. Casts and Unions

Figure 8 shows the data relating to the type safety for these code samples and applications. As shown in the figure, unchecked casts are common in even modern C++ applications. Likewise, unions are found in almost all of these applications. Fixing these issues may be difficult because switching from C-style casts to `static_cast` and `dynamic_cast` requires careful consideration of the inheritance hierarchies and the exact type conversions that are required. In Ironclad C++, the authors noted that extracting inheritance hierarchies was one of the manual tasks that had to be performed by a programmer to conform to the Ironclad C++ safe subset.

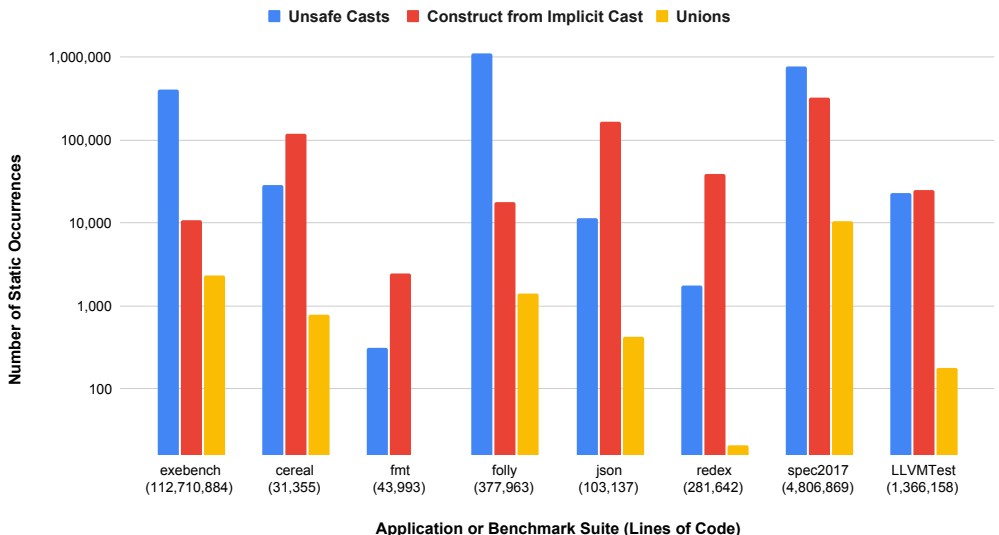

**Figure 8.** Casts and unions found in Exebench and other C++ applications.

We also find that constructors that take the result of an implicit cast are relatively common in all applications examined in this study. Therefore, it may take additional effort to translate to a safe subset if smart pointers are used more frequently within that subset.

The `folly` application is a significant outlier for unsafe casts because it deals with file and network interfaces that require casting from untyped bitstreams to usable data types. Figure 9 shows a heat map of the unsafe casts used in `folly`. Unlike its use of unsafe functions, the unsafe casts in `folly` are concentrated to fewer files and fewer locations in those files.

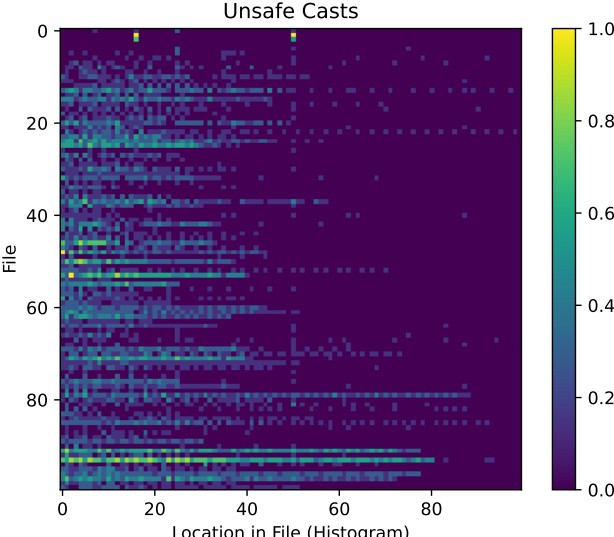

**Figure 9.** Heat map of the unsafe casts in `folly`.

*6.4. References*

Figure 10 shows the total number of reference types used in the contexts of class members, return values, and references initialized by the dereference of a pointer. We did not include the results for the Exebench suite because we found a negligible number of references used in the function samples. Only a small number of Exebench code samples used return by reference. On the other hand, the C/C++ benchmarks and modern C++ applications examined in this study used references in all of these contexts. We note that not all references are dangerous, and return by reference may be acceptable if the returned value does not live beyond its scope. More work will be required on our static analysis tool to narrow down the cases in which returning by reference may be dangerous, and this is based on the rules presented in the existing C++ standards.

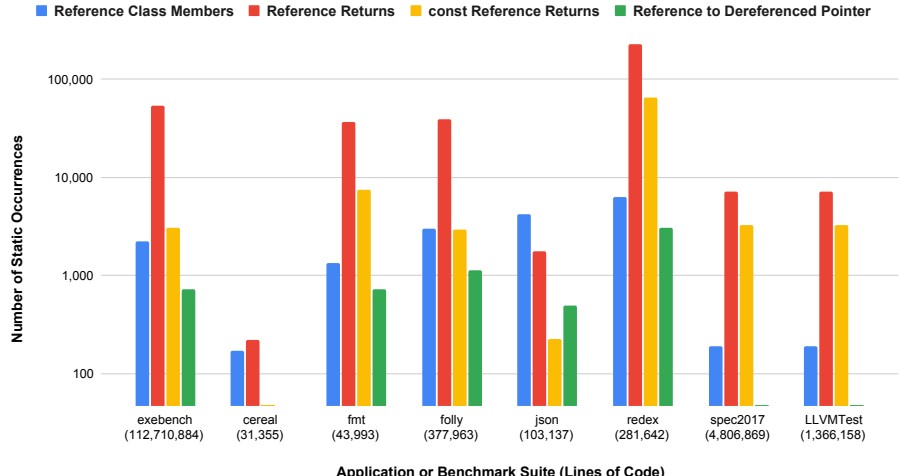

**Figure 10.** References found in Exebench and other C++ applications.

Reference class members and references to dereferenced pointers create the possibility of invalid references due to initialization or lifetime errors unless the initialization is checked. It is difficult to check for initialization in these cases, and more work is required to determine if these cases are dangerous or not. In general, references are commonplace enough in modern C++ code that a solution will be required to ensure memory safety for references in a safe subset of C++.

### 6.5. Arrays

Figure 11 shows the total number of potentially problematic array operations in the C++ samples and applications. In total, the arrays were not nearly as prevalent in the Exebench and modern C++ applications as the pointers and references were. However, there was a high incidence of array operation in the LLVMTest and SPEC 2017 benchmark suites due to the use of arrays in the computation-intensive applications that are common in benchmark suites. Some work may be required to ensure the metadata are maintained and to prevent arrays from decaying to pointers. Further, performance optimizations may be required for two-dimensional and higher arrays. However, the effort required to deal with issues related to arrays seems small compared to handling pointers and references.

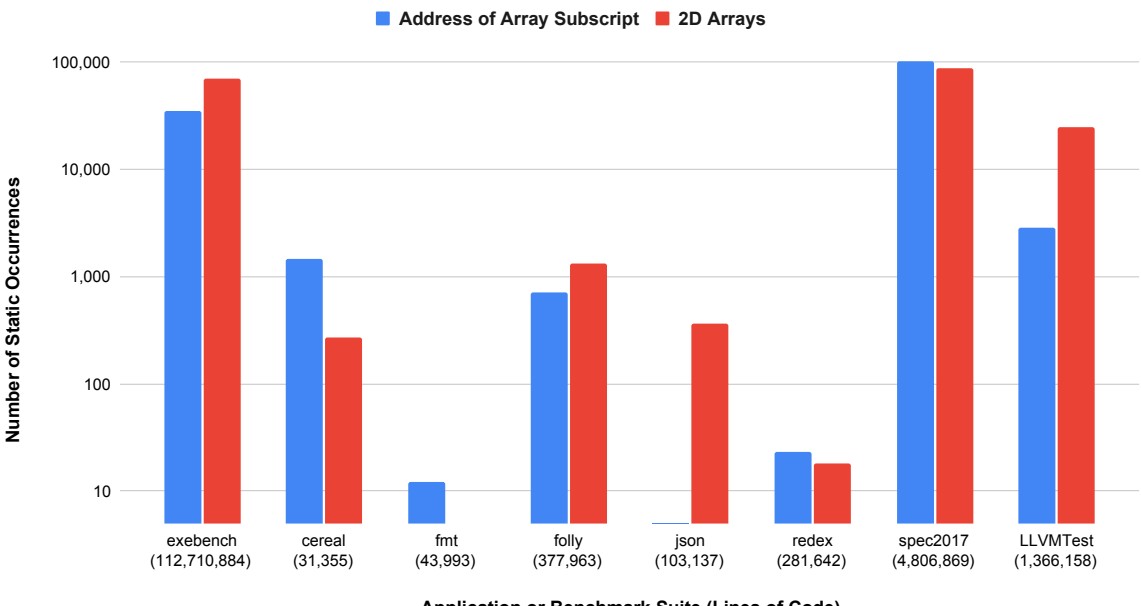

**Figure 11.** Arrays found in Exebench and other C++ applications.

### 6.6. Summary

To summarize, we return to the three questions posed at the beginning of the methodology. For (Q1), we find that pointers are common both in C/C++ code at large and in more modern C++ code. Although modern C++ code has begun to adopt smart pointers, raw pointers, and even void pointers, they are much more common than smart pointers. Significant work, either at the source level or within the compilation tool chain, will be required to replace raw pointers in a safe subset of C++.

For (Q2), we find that problematic constructs like unsafe functions, unsafe casts, and unions are found in both C/C++ code at large and modern C++ code. Again, work will be required to transition from existing C/C++ code toward a safe subset of C++ that avoids these unsafe constructs.

For (Q3), we find that modern C++ code is not all that different in terms of the use of unsafe constructs than C/C++ code at large. The one main difference appears to be the use of references, as we could find little to no use of references in the Exebench code samples.

## 7. Limitations and Future Work

Due to limited time, we did not perform an exhaustive search for all code characteristics related to existing safe C/C++ standards and subsets. We intend to perform a more detailed study of the characteristics found in safe standards on the same set of applications to further classify how well modern C++ code conforms to these standards.

Given the data-driven approach in this paper, we cannot completely verify that the statistics gathered by our static analysis tool are precise. All of the static analysis performed in this paper identifies "trivial" (i.e., not semantic) code properties that should be unaffected by the usual false positives that plague static analysis tools. The main threat to validity is false negatives due to ASTMatcher patterns that may not handle all the possible combinations of C/C++ constructs that can occur in code. We tested the tool on a set of unit tests that demonstrated the code characteristics we wanted to find, but corner cases may exist in the diverse set of C++ code samples we examined in this study. However, it is possible that the tool is under counting these constructs due to unseen cases. In general, it will be impractical to verify that the statistics are exactly correct on all 5.8 million of the samples from Exebench, the C/C++ applications in the two benchmark suites, and a large collection of the C++ files from the modern applications.

## 8. Conclusions

We analyzed the code characteristics of 5.8 million code samples from the Exebench benchmark suite, the LLVMTest and SPEC 2017 benchmark suites, and five modern C++ applications using a static analysis tool. Our analysis of C/C++ code, both at large and in the context of more modern C++ practices, has revealed important insights. We have found that raw pointers remain prevalent in both categories, despite some adoption of smart pointers in modern C++ code. This indicates a substantial need for efforts, whether at the source code level or within the compilation tool chain, to replace raw pointers with safer alternatives in a safe subset of C++.

Furthermore, we find that problematic constructs, such as unsafe functions, unsafe casts, and unions, have shown their persistence in both C/C++ code at large and modern C++ code. This underscores the necessity of a transition from existing C/C++ code toward a safer subset of C++ that avoids these hazardous constructs.

Lastly, we observed that modern C++ code does not significantly differ in terms of safety when compared to C/C++ code at large. The primary distinguishing factor is the usage of references, which are notably absent or rarely utilized in the Exebench code samples. In summary, these findings emphasize the importance of ongoing efforts to enhance the safety and modernization of C/C++ code-bases.

**Funding:** This research received no external funding.

**Data Availability Statement:** All data are publicly available on the Github repository: https://github.com/crdelozier/subsets (accessed on 22 December 2023).

**Acknowledgments:** Thank you to Samidha Nageshwar for her contribution of working on the Python scripts for this project during her time in the SEAP program.

**Conflicts of Interest:** The author declares no conflicts of interest.

## Abbreviations

The following abbreviations are used in this manuscript:

| | |
|---|---|
| AST | Abstract Syntax Tree |
| AUTOSAR | Automotive Open System Architecture |
| CERT | Computer Emergency Response Team |
| HIC | High Integrity C++ |
| JSON | JavaScript Object Notation |
| MISRA | Motor Industry Software Reliability Association |
| JSF | Joint Strike Fighter |

| NSA | National Security Agency (United States) |
|-----|------------------------------------------|
| NIST | National Institute for Standards and Technology (United States) |

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
