# Peer review of "How Close Is Existing C/C++ Code to a Safe Subset?"

_jcp, doi:10.3390/jcp4010001_

Round 1

Reviewer 1 Report

Comments and Suggestions for Authors

Paper summary: This paper analyses a massive number of C++ programs (5.8 million code samples + 6 modern C++ programs) to answer three research questions: (Q1) How often are pointers found in modern C++ code? (Q2) How often are "problematic" code constructs found in modern C++ code? (Q3) Is "modern" C++ code closer to a safe subset than C++ code at large? To answer those questions, the authors built a clang-based static analyzer that examines C++ programs for the use of 1) pointers, 2) unsafe functions such as malloc() and free(), 3) casts, 4) references, and 5) arrays. The results show that such unsafe constructs are still used even in modern C++ programs.

Detailed comments: this paper addresses a very important and imminent problem - i.e., the safety of C/C++ programs. I think this paper is well motivated, appropriately discusses unsafe constructs by reviewing various safe coding standards such as MISRA C++, and effectively analyzes a massive number of C++ programs. While the results are not very surprising, I'd say this paper will up a new research opportunity to make C/C++ safer. That's the reason for the recommendation. While I think the paper is already well-written, here are my suggestions for improving the paper:

1. The static analyzer
I'd suggest discussing how accurate the static analyzer is. Specifically, could the results include true/false positive/negative due to the nature of static analysis?

For example, the following sentences made me wonder if other user-defined smart pointers than "ptr" are used in the applications:
> L430: Smart Pointers are identified by searching for declarations with a C++ class type with the names unique_ptr, shared_ptr, weak_ptr, auto_ptr, and ptr. We note that ptr is not a standard library smart pointer type, but at least one of the applications that we examined created their own smart pointer wrappers with this name.

Also, this sentence somewhat implies that the analyzer is not perfect:
> L596: We do not include results for the Exebench suite because we found a negligible number of references used in the function samples.

I don't think those will significantly change the overall results, but discussing this in Section 6 is still very important.

2. Interpreting/Analyzing the results
While measuring the number of occurrences is a good starting point to capture rough trends, it'd be great to somehow associate it with code coverage results. For example, are those unsafe patterns scattered across the entire code? Or would there be any trend where some patterns are only found in a specific portion of the code? Ideally, it'd be great if the analyzer could work with a code coverage tool, but it'd still be useful to create a histogram where the x-axis shows each code location (clang::SourceLocation) and the y-axis shows the number of occurrences based on the output from the static analyzer.

Minor comments:
- Please add a link to each Modern C++ application (cereal, fmt, folly, json, and redex).
- L592: TODO: what’s up with folly?

Author Response

Thank you for your time and feedback.  Your comments have been beneficial in helping to revise and improve this paper.

I'd suggest discussing how accurate the static analyzer is. Specifically, could the results include true/false positive/negative due to the nature of static analysis?

All of the static analysis performed in this paper identifies "trivial" (i.e. not semantic) code properties that should be unaffected by the usual false positives that plague static analysis tools.  The main threat to validity is false negatives due to ASTMatcher patterns that may not handle all possible combinations of C/C++ constructs that can occur in code.  I have attempted to avoid false negatives by testing the static analysis tool on a suite of unit tests that demonstrate each of the patterns.  However, it is possible that the tool is undercounting these constructs due to unseen cases.  As far as I know, the only way to double-check the results would be to count by hand or have a third party reimplement the tool, which is impractical.  I have added a more thorough discussion to the Limitations and Future Work section.

While measuring the number of occurrences is a good starting point to capture rough trends, it'd be great to somehow associate it with code coverage results. For example, are those unsafe patterns scattered across the entire code? Or would there be any trend where some patterns are only found in a specific portion of the code? Ideally, it'd be great if the analyzer could work with a code coverage tool, but it'd still be useful to create a histogram where the x-axis shows each code location (clang::SourceLocation) and the y-axis shows the number of occurrences based on the output from the static analyzer.

Thank you for this excellent suggestion!  I have added a heat map of the occurrences of unsafe functions and unsafe casts in folly to demonstrate how frequently these patterns occur in the code.  For the sake of the length of the paper, I only added two heat maps, but I could certainly add more in supplementary material if the addition of results beyond these two examples would be valuable.

Minor comments

I have added links to each of the C++ applications and fixed the TODO.  Thank you for catching both of these issues!

Reviewer 2 Report

Comments and Suggestions for Authors

I have the following comments and suggestions for the author.

Overall, it is not entirely clear what language, C or C++ or both, is the subject of this work.

Namely, the majority of the paper is concerned with C++: Section 2 gives a very thorough description of different standards for safe C++, Section 4 outlines the methodology centred around the safe C++ rules. At the same time, Section 3 mostly concentrates around enforcing safety in C, and Sections 2 and 4 barely mention any standards for safe C.

But most importantly, the proposed methodology is inconsistent with the experimental evaluation. In particular, the usage of Exebench in this work does not seem appropriate (or at least it does not align with the methodology). According to the description of the dataset provided in this paper (specifically, line 512) and the description given on GitHub, Exebench is a collection of executable C (not C++) functions. And unsurprisingly, the results show that it is full of features that are deemed "unsafe" in C++ (i.e., Clib functions, malloc, free, absence of smart pointers, C-style casts, etc.). As a result, this analysis is pointless in the context of safe C++ despite the large number of programs in the dataset.

I suggest 1) reducing the scope of the paper from C/C++ to just C++, 2) removing the evaluation of Exebench from this work, and 3) expanding the evaluation section with more real-world open-source C++ programs.

Comments on the Quality of English Language

- Several typos (e.g., "C+" instead of "C++" in lines 142, 173)

- Unaddressed TODO note in line 592

Author Response

Thank you for your time and feedback.  Your comments have been beneficial in helping to revise and improve this paper.

Overall, it is not entirely clear what language, C or C++ or both, is the subject of this work.  I suggest reducing the scope of the paper from C/C++ to just C++.

I firmly believe that it is important to examine both C and C++ in the context of moving towards using a safe subset of C++.  However, after reading your comments, I agree that the paper was not clear about the distinction between C and C++ code and the potential to secure code from both languages using a safe subset of C++.

I have revised the paper in the following ways.

  • Added a Summary subsection to Section 3 to discuss the distinction between C and C++ with regard to safe subsets of C/C++
  • Added a Language column to Table 2 to clarify which language is predominant
  • Revised Results and Conclusions to differentiate between C code, C++ code, and "modern" C++ code where appropriate

In a discussion of a safe subset of C++, the goal of securing C, C++, or a mix of the two languages must be considered.  In my view, it is difficult to separate C++ from the unsafe features of its predecessor, including raw pointers, arrays, unchecked casts, and unsafe functions from the C library.  Therefore, I examine applications that are predominantly C, a mix of C and C++, and predominantly C++ in this study to fully understand the effort required to refactor these types of applications to a safe subset of C++.

According to the description of the dataset provided in this paper (specifically, line 512) and the description given on GitHub, Exebench is a collection of executable C (not C++) functions. 

Exebench contains some C++ code constructs, despite being described as a collection of C functions.  However, it is more predominantly C code than C++.  I have corrected this description in the paper and revised the paper to more completely discuss the differences between C code, C/C++ code, and modern C++ code.

I suggest expanding the evaluation section with more real-world open-source C++ programs.

Thank you for this suggestion!  In addition to differentiating between C code, C/C++ code, and modern C++ code in the paper, I have also added two new benchmark suites to the paper.  Both the LLVM test suite and the SPEC 2017 benchmark suite contain a mix of C and C++ applications.  I hope that the additional data will bridge the gap between the largely C code in Exebench to the modern C++ applications.

Reviewer 3 Report

Comments and Suggestions for Authors

Paper summary                                                                                                   

In this paper, the author conducts an analysis of code characteristics from 5.8 million code samples using static analysis tools. By examining the frequency and prevalence of pointers, unsafe functions, casts and unions, references and arrays widely employed in large and modern C++ applications, they conclude that the overall disparity between current C++ code and modern C++ code is not substantial. A significant amount of work is still required for a smooth transition from existing C/C++ code to a secure subset of C++.

Strengths

+ The research content is meaningful as examine how close existing C/C++ code is to conforming to a safe subset of C++.

+ The starting point of the paper design is novel

+ The experiment is relatively complete and the results are informative for subsequent studies.

+ The dataset used in the experiment is both authentic and comprehensive.

Weaknesses

The description of method is not clear.

Unconvincing experimental results.

Lack of innovation.

***Method***

Among the three main contributions in this paper, the author notes one contribution, which is the proposal of a static tool for identifying potential issues. However, the description of the use of the static tool and the method is only mentioned in lines 411-417, without providing detailed explanations of the specific design principles of the method. Providing a more detailed explanation of the method steps would help readers better understand the design of this method.

***Experiment setting***

The passage mentions, "In general, it will be almost impossible to verify whether the statistical data on all 5.8 million samples is entirely accurate." Given that the primary focus of this paper is the statistical analysis of the frequency and commonality of unsafe constructions in existing C/C++ code, leading to the conclusion of "How close is existing C/C++ code to a safe subset?", the accuracy of the statistical data is a paramount consideration. It directly affects the reliability of the paper's analytical results. Therefore, could there be an experimental addition to test the accuracy of the statistics further and enhance the credibility of the findings?

***innovativeness***

This paper lacks some degree of innovativeness. While it is pioneering in its use of static analysis tools to determine the proximity between existing C/C++ code and a secure subset of C++, the specific static analysis tool employed was not developed by the researchers themselves. Instead, they referenced the existing clang's ASTMatchers library for the analysis. During the experimental phase, the researchers solely utilized this static analysis tool to summarize and analyze the frequency of the use of pointers and unsafe functions in the dataset. Despite the substantial workload involved, the paper falls slightly short in terms of innovativeness.

Comments on the Quality of English Language

The article is well-structured, with clear sectioning and logical flow, making it easy to follow and understand.

Author Response

Thank you for your time and feedback.  Your comments have been beneficial in helping to revise and improve this paper.

Therefore, could there be an experimental addition to test the accuracy of the statistics further and enhance the credibility of the findings?

All of the static analysis performed in this paper identifies "trivial" (i.e. not semantic) code properties that should be unaffected by the usual false positives that plague static analysis tools.  The main threat to validity is false negatives due to ASTMatcher patterns that may not handle all possible combinations of C/C++ constructs that can occur in code.  I have attempted to avoid false negatives by testing the static analysis tool on a suite of unit tests that demonstrate each of the patterns.  However, it is possible that the tool is undercounting these constructs due to unseen cases.  As far as I know, the only way to double-check the results would be to count by hand or have a third party reimplement the tool, which is impractical.  I have added a more thorough discussion to the Limitations and Future Work section.

The specific static analysis tool employed was not developed by the researchers themselves. Instead, they referenced the existing clang's ASTMatchers library for the analysis.  During the experimental phase, the researchers solely utilized this static analysis tool to summarize and analyze the frequency of the use of pointers and unsafe functions in the dataset. Despite the substantial workload involved, the paper falls slightly short in terms of innovativeness.

Although the static analysis tool is based on the ASTMatchers library, the specific tool (e.g. matcher patterns) was developed by the researchers for this study.  Identifying these specific code features based on the prior work on safe standards and safe subsets is an innovative part of the work, even if the static analysis techniques use existing tools.  I have added a figure to the manuscript to demonstrate the tool's use of the ASTMatchers library.

Round 2

Reviewer 2 Report

Comments and Suggestions for Authors

I like the revised version of the paper. Most of my major comments were addressed by the author. Saying that, I have a few minor suggestions:

- The phrase "safe subset" that is used throughout the paper sounds a bit too general. It does not appear to be something well established (i.e., I didn't notice a reference to it in other works), but rather something that the author proposes in this paper. So I recommend making it clear from the start (abstract and introduction); perhaps something along the lines of: "we propose a subset of safety requirements/features/recommendations (see Section 4.1) and evaluate a set of C/C++ benchmarks against this subset".

- Great discussion in Section 3.6. The last paragraph provides a good justification to talk about both C and C++ in this paper. So I think it would be beneficial to mention this in the early introduction to set the motivation for the rest of the paper.

- All figures featuring bar plots are missing axes labels (this should be addressed). Also, the author could add the number of lines of code in each evaluated software (perhaps just right under the name) to the plots that present occurrences of different unsafe features.

Author Response

Thank you again for your time and feedback on this paper!  I appreciate your help with improving the manuscript.

The phrase "safe subset" that is used throughout the paper sounds a bit too general. It does not appear to be something well established (i.e., I didn't notice a reference to it in other works), but rather something that the author proposes in this paper. So I recommend making it clear from the start (abstract and introduction); perhaps something along the lines of: "we propose a subset of safety requirements (see Section 4.1) and evaluate a set of C/C++ benchmarks against this subset".

I have added a definition of a safe subset with references to prior work in the introduction.  Prior work has used different terms for a safe subset, including a safe dialect and type-safe retrofitting, but the idea is the same.  The following text has been added to the introduction.

"A safe subset of C/C++ restricts or bans the use of unsafe language features and, when necessary for productivity or performance, replaces unsafe language features with safer variants that accomplish the same goal.  Prior work \cite{safec, ccured, cyclone, ironclad} has aimed to design a safe subset of C/C++.  These efforts have generally restricted the use of types and operations that may allow unsafe accesses to memory when used incorrectly, and they have statically restricted the use of features outside of the safe subset via a compiler or static analyzer."

Great discussion in Section 3.6. The last paragraph provides a good justification to talk about both C and C++ in this paper. So I think it would be beneficial to mention this in the early introduction to set the motivation for the rest of the paper.

I have added multiple sentences to the introduction to better motivate discussion of C and C++ in this paper.  The new text is included below.

"C allows unchecked access to memory through pointers and arrays, and C++, its predecessor, maintains the same capabilities.  These features make C and C++ powerful languages, giving programmers direct control over memory layout, allocation, and access.  However, programmers often make mistakes involving memory, and these mistakes can lead to security vulnerabilities."

"We examine code that is predominantly C, predominantly C++, and a mix of both to investigate how the work required to conform to a safe subset might vary between code bases."

All figures featuring bar plots are missing axes labels (this should be addressed). Also, the author could add the number of lines of code in each evaluated software (perhaps just right under the name) to the plots that present occurrences of different unsafe features.

I have made the requested edits to the bar plots.

Reviewer 3 Report

Comments and Suggestions for Authors

All my concerns have been addressed in the authors' revised version. I think it can be accepted.

Author Response

Thank you for your time and feedback in improving this paper!